# Protocatechuic Aldehyde Inhibits α-MSH-Induced Melanogenesis in B16F10 Melanoma Cells via PKA/CREB-Associated MITF Downregulation

**DOI:** 10.3390/ijms22083861

**Published:** 2021-04-08

**Authors:** Seok-Chun Ko, Seung-Hong Lee

**Affiliations:** 1Department of Genetic Resources, National Marine Biodiversity Institute of Korea, Seocheon 33662, Korea; seokchunk@mabik.re.kr; 2Department of Pharmaceutical Engineering and Medical Science, Soonchunhyang University, Asan 31538, Korea

**Keywords:** protocatechuic aldehyde, anti-melanogenesis activity, molecular mechanisms, melanoma cells

## Abstract

Protocatechuic aldehyde (PA) is a naturally occurring phenolic compound that is a potent inhibitor of mushroom tyrosinase. However, the molecular mechanisms of the anti-melanogenesis activity of PA have not yet been reported. The aim of the current study was to clarify the melanogenesis inhibitory effects of PA and its molecular mechanisms in murine melanoma cells (B16F10). We first predicted the 3D structure of tyrosinase and used a molecular docking algorithm to simulate binding between tyrosinase and PA. These molecular modeling studies calculated a binding energy of −527.42 kcal/mol and indicated that PA interacts with Cu400 and 401, Val283, and His263. Furthermore, PA significantly decreased α-MSH-induced intracellular tyrosinase activity and melanin content in a dose-dependent manner. PA also inhibited key melanogenic proteins such as tyrosinase, tyrosinase-related protein 1 (TRP-1), and TRP-2 in α-MSH-stimulated B16F10 cells. In addition, PA decreased MITF expression levels by inhibiting phosphorylation of cAMP response element-binding protein (CREB) and cAMP-dependent protein kinase A (PKA). These results demonstrate that PA can effectively suppress melanin synthesis in melanoma cells. Taken together, our results show that PA could serve as a potential inhibitor of melanogenesis, and hence could be explored as a possible skin-lightening agent.

## 1. Introduction

Melanin is produced by melanocytes distributed in the basal layer of the epidermis, and is a key element of skin, hair, and eye color. Melanin is produced through a series of complex physiological processes that include enzymes and chemical reactions involved in melanogenesis. Three enzymes, tyrosinase, tyrosinase-related protein-1 (TRP-1), and TRP-2, play critical roles in melanogenesis [1,2]. Microphthalmia-associated transcription factor (MITF) is an essential factor that regulates the transcription of the key melanogenic proteins [3]. Activation of both the cAMP-dependent protein kinase A (PKA) and cAMP response element-binding protein (CREB) signaling pathways is known to play an essential role in MITF expression and activity. Phosphorylation of CREB is activated by phosphorylation of PKA, leading to expression of MITF and resulting in an increase in melanin synthesis [3,4]. Thus, downregulation of PKA and CREB signaling in melanocytes inhibits melanin synthesis by downregulating MITF expression.

Under normal physiological conditions, melanin production is essential in protecting DNA and skin from the effects of UV radiation. However, excessive melanin production (hyperpigmentation) causes various dermatological disorders, such as melasma, freckles, age spots, and malignant melanoma [5]. In addition, Asian women in particular prefer white skin to dark skin [6]. Thus, whitening agents or anti-melanogenesis factors are being used to solve the problem of hyperpigmentation. Aside from these functions, whitening agents have also been employed to deal with dark spots and uneven skin color problems to promote uniformity of skin color, which many people feel is synonymous with health or a beautiful appearance [7,8,9]. Hence, skin whitening products, such as lightening creams or whitening lotions, are a growing segment of the global cosmetic and beauty industry, which is expected to experience a dramatic increase and be worth an estimated $US 31.2 billion by 2024 [10].

Several known melanogenesis inhibitors, such as kojic acid and arbutin, are used in cosmetics and medicines to treat pigment abnormalities. However, these inhibitors come with various safety concerns and low whitening bioactivity. Thus, a number of studies are currently attempting to develop naturally derived melanogenesis inhibitors without undesirable side effects. Protocatechuic aldehyde (PA) is a naturally occurring phenolic compound that can be found in many plants and organisms [11,12,13,14]. Previous studies have reported that PA possesses a potent inhibitory effect against mushroom tyrosinase and suggested that it is a potential agent for treatment of pigmentation disorders [15]. However, the precise anti-melanogenesis activity of PA and the molecular mechanisms involved in this activity at the cellular level have not yet been reported. Therefore, the aim of the current study was to further investigate the anti-melanogenesis activity and mechanism of action of PA, a potential melanogenesis inhibitor, by analyzing tyrosinase binding through molecular docking studies and examining its effect on melanogenesis in α-MSH-induced B16F10 melanoma cells.

## 2. Results

### 2.1. In Silico Molecular Docking of Tyrosinase

Docking studies were performed to gain insight into the probable binding conformation of PA and in order to compare PA binding with binding of arbutin, a commercial tyrosinase inhibitor. Docking of the tyrosinase-ligand complexes was performed with PA or arbutin stably bound to tyrosinase in Discovery Studio (DS) 3.0. For arbutin, the 2D diagram indicates binding interactions with the following amino acid residues at the tyrosinase enzyme binding site: Val283 (Pi-alkyl interaction and hydrogen bond), His263 (Pi-Pi stacked interaction), and Gly281 (hydrogen bond) (Figure 1B). PA interacts with the Val283 (Pi-alkyl interaction) and His263 (Pi-Pi stacked interaction) amino acid residues as well as copper ions (Cu400 and 401) through metal–acceptor interactions at the tyrosinase enzyme binding site (Figure 1C). Moreover, the binding energies of PA and arbutin obtained from the DS 3.0 binding energy program were −527.42 and −117.84 kcal/mol, respectively. These results suggest that the tyrosinase binding energy of PA is more stable and stronger than that of arbutin, a commercial tyrosinase inhibitor.

### 2.2. Cytotoxic Effects of PA on B16F10 Melanoma Cells

We first evaluated the cytotoxic effects of PA on B16F10 melanoma cells treated with PA at various concentrations for 72 h via MTT assay. As shown in Figure 2A, PA did not exhibit a cytotoxic effect on B16F10 cells at concentrations ranging from 0.5 to 2 µg/mL. Thus, PA concentrations of 0.5–2 µg/mL were selected for further experiments. Although cytotoxicity was observed at a concentration of 4 µg/mL, a previous study reported that PA did not significantly affect cellular viability at concentrations above 4 µg/mL in another cell line [13]. Moreover, protective effects against toxic substances were also reported [16].

### 2.3. Effects of PA on Intracellular Tyrosinase Activity and Melanin Synthesis in α-MSH-Induced B16F10 Melanoma Cells

To verify the melanogenesis inhibitory effect of PA, we measured the intracellular melanin quantity and tyrosinase activity in α-MSH-induced B16F10 cells. As shown in Figure 2B, PA significantly decreased the α-MSH-induced intracellular melanin content in a dose-dependent manner (vs. cells treated with α-MSH alone). Moreover, PA was more effective than arbutin, a well-known melanogenesis inhibitor, even at a low concentration (2 µg/mL). Tyrosinase is an important enzyme in mammalian melanin synthesis. We measured the tyrosinase inhibitory effects of PA to determine whether PA directly affects intracellular tyrosinase activity. As shown in Figure 2C, when B16F10 cells were treated with α-MSH, tyrosinase activity was significantly increased in comparison with that seen in the absence of α-MSH (control). However, PA dose-dependently decreased cellular tyrosinase activity compared to treatment with α-MSH alone. PA had a similar inhibitory effect against tyrosinase activity as arbutin at low concentrations, which is evidence of a stronger inhibitory effect than that of arbutin. These results suggest that PA inhibits tyrosinase activity and that this inhibitory effect may lead to a decrease in cellular melanin synthesis in B16F10 cells.

### 2.4. Effects of PA on the Expression Levels of MITF, Tyrosinase, TRP-1, and TRP-2 in α-MSH-Induced B16F10 Cells

To explore the molecular mechanisms underlying the anti-melanogenic effect of PA, we determined the effects of PA on the expression levels of the melanogenesis-related signaling molecules MITF, tyrosinase, TRP-1, and TRP-2 proteins using Western blot analysis. MITF is a critical transcription factor that plays a critical role in melanogenesis by activating key melanogenic proteins such as tyrosinase, TRP-1, and TRP-2. Therefore, we first investigated the expression of MITF protein after PA treatment. As indicated in Figure 3, MITF expression was inhibited in α-MSH-induced B16F10 cells by treatment with PA in a dose-dependent manner (vs. α-MSH treatment alone). PA also effectively reduced tyrosinase, TRP-1, and TRP-2 protein expression levels in α-MSH-induced B16F10 cells, as shown in Figure 3. These results suggest that inhibition of melanogenesis by PA is associated with inhibition of melanin synthesis through downregulation of melanogenesis-related protein expression.

### 2.5. Effect of PA on CREB and PKA Expressions in α-MSH-Induced B16F10 Cells

Phosphorylation of CREB and PKA stimulates expression of MITF, leading to an increase in melanin synthesis in melanocytes. To further understand the mechanisms involved in the regulation of MITF expression by PA, we next investigated whether PA could influence CREB/PKA-mediated signaling pathways in α-MSH-stimulated B16F10 cells. As shown in Figure 4, the phosphorylation levels of PKA and CREB were increased in α-MSH-induced B16F10 cells. However, the phosphorylation levels of PKA and CREB were decreased in PA-treated α-MSH-stimulated B16F10 cells in a dose-dependent manner. These results indicate that PA suppresses expression of MITF through downregulation of PKA and CREB signaling in melanocytes.

## 3. Discussion

Previous studies have reported that PA exerts a potent inhibitory effect against mushroom tyrosinase, suggesting the potential of PA for suppression of melanogenesis [15]. Given these results, we set out to elucidate the anti-melanogenesis effects of PA and the molecular mechanisms involved in this activity at the cellular level.

In in silico molecular docking, molecular modeling techniques are used to predict how a protein (enzyme) interacts with small molecules [17,18,19]. The ability of a protein to interact with small molecules plays a major role in the dynamics of that protein, and may enhance or inhibit its biological function. Human tyrosinase has a significant role in melanin biosynthesis in the human body. Therefore, human tyrosinase is a potent target enzyme in the design of novel inhibitors for hyperpigmentation. However, the mushroom tyrosinase of *Agaricus bisporus* was used in this study docking analysis as the 3D crystal structure of human tyrosinase is not available in PDB. Tyrosinase from *Agaricus bisporus* is a major enzyme with high similarity and homology to human tyrosinase [20,21]. In addition, among non-mammalian tyrosinases, the tyrosinase from *Agaricus bisporus* has been studied extensively as a model system for screening of tyrosinase inhibitors and melanogenic studies due to its good properties, namely its structural, functional, and biochemical characteristics [20,22]. For this reason, we also selected tyrosinase from *Agaricus bisporus* (PDB code: 2Y9X) because it is the most suitable for the examination of a model system for screening of tyrosinase inhibitors among non-mammalian tyrosinase. In the present study, we first performed a study of PA docking at the active site of mushroom tyrosinase to understand the mechanism underlying the interaction between tyrosinase and PA. These docking studies provided important information about the operation of the inhibitor that interacted with the tyrosinase binding pocket. According to the predicted conformation of PA in the enzyme binding site, the hydroxyl group forms a metal complex with copper ions. In addition, pi–pi stacked interactions between PA and His263 were observed. Pi-alkyl interactions were detected between PA and Val283 amino acid residues. The results of the present study suggest that interactions, especially the interactions between the hydroxyl group of PA and copper ions, could inhibit tyrosinase activity. The particular amino acid residues that were found to play important roles in the inhibitory activity of PA according to the docking results are in agreement with the literature [23]. It has been suggested that the presence of a hydroxyl group and of an electron donator group in the phenol ring is a primary requirement to effectively serve as a tyrosinase substrate [24]. In the present study, PA was a more effective inhibitor of tyrosinase than arbutin, implying that the number of hydroxyl groups attached to the benzene ring has an important role in tyrosinase inhibition activity. However, it should be cautioned that tyrosinase from mushroom is different from human tyrosinase due to the presence of additional amino acids in the hydrophobic substrate-binding pocket of mushroom tyrosinase [22]. Therefore, recently, a rather large number of studies have been directed to the utilization of human tyrosinase for the search and assessment of tyrosinase inhibitors [25,26]. Although further human tyrosinase studies are required to clarify the efficacy of PA in a tyrosinase activity inhibition, the present results suggest that PA could serve as a potential inhibitor of tyrosinase.

Tyrosinase plays a crucial role as a rate-limiting enzyme that controls melanin production. An increase in melanin production is directly correlated with tyrosinase activity [27]. Therefore, we investigated whether PA could inhibit melanin synthesis through inhibition of tyrosinase activity in α-MSH-induced melanoma cells. We found that PA significantly inhibited melanin synthesis and tyrosinase activity in a dose-dependent manner, and did not have a cytotoxic effect on melanoma cells. Accordingly, the present study suggests that PA may effectively downregulate tyrosinase activity in α-MSH-induced melanoma cells, resulting in a decrease in cellular melanin synthesis. In addition, both the previous and current results confirm that PA not only inhibits cell-free tyrosinase activity but also inhibits intracellular tyrosinase in α-MSH-induced melanoma cells. L-tyrosine and L-DOPA as the consecutive key substrates and intermediates of melanogenic pathways play important roles as positive regulators of melanogenesis [28]. In the melanogenic initial pathway, tyrosinase plays a central role in melanin biosynthesis by catalyzing the hydroxylation of tyrosine to form L-DOPA, and from L-DOPA to DOPA quinone [28,29,30]. Previous studies [15] and the present results showed that PA had a strong inhibitory effect in tyrosinase activity inhibition experiments using dopa and tyrosine as substrates, respectively. Therefore, the results of previous and present studies clearly demonstrated that PA could counteract stimulation of melanogenesis by both L-tyrosine and L-DOPA. In addition, substrates for tyrosinase are generally either phenols, which undergo monooxygenation, or catechols, which undergo oxidation, leading in both cases to orthoquinones [31]. However, it has previously been reported that some polyphenolic compounds with an electron-withdrawing group attached directly to the benzene, such as hydroquinone and catechols, do not act as substrates in mushroom tyrosinase [21]. Thus, we believe that PA, a naturally derived catechol with an electron-withdrawing group attached directly to the benzene ring, does not act as a substrate for tyrosinase but might act as an inhibitor, as shown in the present study. Furthermore, it is necessary to further examine the efficacy of PA by checking whether or not the PA acts as a substrate of human tyrosinase being modified on exposure to the human tyrosinase.

MITF is a key regulator of melanogenesis [3,32]. Therefore, it is essential to measure the extent of downregulation of MITF expression to assess the melanogenesis inhibitory effect of PA. In the present study, PA significantly suppressed the expression of MITF in α-MSH-induced melanoma cells. MITF expression can stimulate tyrosinase, TRP-1, and TRP-2 expression, thus increasing melanin production [33]. We confirmed that PA also significantly suppressed the expression of tyrosinase, TRP1, and TRP2. The results of the present study demonstrated that PA inhibits MITF expression and results in downregulation of tyrosinase, TRP1, and TRP2 levels, ultimately leading to decreased melanin production.

It has been reported that secretion of α-MSH due to UV exposure can increase PKA activity [34]. Activation of PKA subsequently upregulates CREB phosphorylation, which positively regulates the expression of MITF; thus, the elevation of MITF expression is ultimately stimulated by CREB phosphorylation [35]. To illuminate the mechanisms involved in the PA-induced reduction of MITF expression, we evaluated the protein levels of PKA and CREB, a key transcription factor that is involved in MITF expression. The results of the present study showed that PA treatment suppresses both PKA and CREB activation, resulting in downregulation of MITF expression. These results suggest that PA can inhibit melanin synthesis by inhibiting MITF through PKA/CREB signaling cascades, subsequently downregulating the expression levels of TYR, TRP-1, and TRP-2 in α-MSH-induced B16F10 melanoma cells.

MITF is recognized as a major regulator of melanoma progression, but it can regulate multiple biological processes in melanoma cells, such as suppression of metastasis, differentiation, proliferation, migration, and senescence [36]. The increased MITF expression is associated with melanoma differentiation and proliferation, whereas decreased MITF expression is associated with a melanoma dedifferentiation [37]. The present study shows that PA downregulated MITF expression. Therefore, these results suggest that PA may affect melanoma dedifferentiation induction through inhibition of MITF expression.

In conclusion, the present results confirm that PA inhibits not only cell-free tyrosinase activity but also intracellular tyrosinase in α-MSH-induced melanoma cells. Additionally, PA was identified as a competitive inhibitor of tyrosinase through a molecular docking study. In addition, the present study shows for the first time that PA inhibits melanogenesis via PKA/CREB-associated MITF downregulation in α-MSH-induced melanoma cells. Although more extensive and further studies, including human tyrosinase and human melanoma cells studies, are needed for a thorough understanding of the potential inhibitor of PA in melanogenesis, our results confirm that PA could be explored as a possible skin-lightening agent.

## 4. Materials and Methods

### 4.1. Materials

Protocatechuic aldehyde (purity; ≥97%, Figure 1A), dimethyl sulfoxide (DMSO), 3-(4,5-dimethylthiazol-2-yl)-2,5-diphenyltetrazolium bromide (MTT), L-DOPA, arbutin, and α-melanocyte stimulating hormone (α-MSH) were purchased from Sigma-Aldrich (St, Louis, MO, USA). Dulbecco’s modified Eagle’s medium (DMEM), phosphate-buffered saline (pH 7.4; PBS), fetal bovine serum (FBS), and penicillin–streptomycin (P/S) were obtained from Gibco BRL (Grand Island, NY, USA). Tyrosinase, TRP1, TRP2, MITF, and GAPDH were obtained from Santa Cruz Biotechnology (Santa Cruz, CA, USA). Here, p-CREB, CREB, p-PKA, and PKA were purchased from Cell Signaling Technology (Beverly, MA, USA). Anti-rabbit IgG secondary antibody was purchased from Thermo Fisher Scientific (Waltham, MA, USA).

### 4.2. In Silico Molecular Docking Analysis of Tyrosinase

In order to assess the interaction between PA and tyrosinase, a molecular docking analysis was carried out using the CDOCKER in Accelrys Discovery Studio (DS) 3.0 (Accelrys, Inc., San Diego, CA, USA). The crystal structure of tyrosinase (PDB code: 2Y9X) [22] was obtained from the Protein Data Bank (PDB, http://www.pdb.org, accessed on 22 January 2021).

### 4.3. Cell Culture

B16F10 mouse melanoma cells were obtained from American Type Culture Collection (ATCC; Manassas, VA, USA). Cells were cultured with Dulbecco’s modified Eagle’s medium supplemented with 10% (*v*/*v*) heat-inactivated fetal bovine serum, 100 µg/mL streptomycin, and 100 U/mL penicillin. Cells were sub-cultured every two days and maintained at 37 °C with 5% CO_2_ in a humidified incubator.

### 4.4. Cell Viability Assessment

Cell viability was quantified through a colorimetric MTT assay [38] by measuring the mitochondrial activity of viable cells. Briefly, B16F10 cells were seeded (5 × 10^4^ cells/mL) into 96-well culture plates and incubated with various concentrations of PA for up to 72 h prior to MTT treatment. An MTT stock solution (50 μL; 2 mg/mL in PBS) was added to each well to achieve a total reaction volume of 250 μL. After 4 h of incubation, the plates were centrifuged at 2000 rpm for 10 min. After aspiration of the supernatants, the formazan crystals in each well were dissolved in DMSO. The amount of purple formazan was assessed by measuring the absorbance at 540 nm using a spectrophotometer (TECAN, Salzburg, Austria).

### 4.5. Assessment of Melanin Contents

Melanin contents were assessed using a published protocol [39] with slight modifications. Briefly, B16F10 cells (5 × 10^4^ cells/mL) were seeded into 24-well plates and incubated in the presence or absence of 100 nM α-MSH. These cells were then incubated with various concentrations of PA for 72 h. The plates were washed with PBS, then 1 N NaOH containing 10% DMSO was added to each well. After incubation at 80 °C for 1 h and mixing to solubilize melanin, the amount of melanin was assessed by measuring the absorbance at 475 nm.

### 4.6. Assessment of Tyrosinase Activity

Tyrosinase activity was measured using a previously described method [40] with minor modifications. Briefly, B16F10 cells were stimulated with α-MSH and treated with various concentrations of PA and arbutin. After incubation at 37 °C for 72 h, the cells were washed with PBS and lysed in 50 nM sodium phosphate buffer (pH 6.8) containing 1% Triton X-100 and 0.1 mM phenylmethylsulfonyl fluoride (PMFS). Cell lysates were harvested by centrifugation at 12,000 rpm for 30 min at 4 °C. After quantification and normalization, the cell lysate was incubated with 10 mM L-DOPA at 37 °C for 1 h to promote dopachrome formation. The solution was then measured using a spectrophotometer at 492 nm.

### 4.7. Western Blot Analysis

After melanoma cells were treated with the indicated concentrations of PA for the indicated times, the cells were washed twice with cold PBS and harvested. Next, the cells were lysed with RIPA lysis buffer and centrifuged at 12,000 rpm for 20 min at 4 °C to harvest the cell lysates. Protein concentrations were determined using a BCA^TM^ protein assay kit (Pierce, Rockford, IL, USA). Samples containing equal protein contents were subjected to 15% sodium dodecyl sulfate–polyacrylamide gel electrophoresis (SDS-PAGE). After separation by electrophoresis, the proteins were transferred into nitrocellulose membranes (Bio-Rad, Hercules, CA, USA). The membranes were blocked with 5% nonfat dry milk in TBST (25 mM Tris–HCl, 137 mM NaCl, 2.65 mM KCl, 0.05% Tween 20, pH 7.4) for 2 h at room temperature and then incubated with each of the following primary antibodies: MITF, tyrosinase, TRP1, TRP2, p-CREB, CREB, p-PKA, PKA, and GAPDH. Thereafter, the membranes were washed with TBST and incubated with secondary antibodies for 2 h at room temperature. Protein bands were visualized using an ECL Western blotting detection kit (Santa Cruz, CA, USA) and captured with a FUSION SOLO Vilber Lourmat system (Vilber, Paris, France).

### 4.8. Statistical Analysis

All experiments were carried out in triplicate. Data are described as means ± standard deviations (SDs). Statistical analysis was done using one-way analysis of variance (ANOVA) complemented by Duncan’s multiple range test. Statistical significance was considered at *p* < 0.05. Various degrees of significance were indicated as follows: *, *p* < 0.05; **, *p* < 0.01; ^##^, *p* < 0.01.

## Figures and Tables

**Figure 1 ijms-22-03861-f001:**
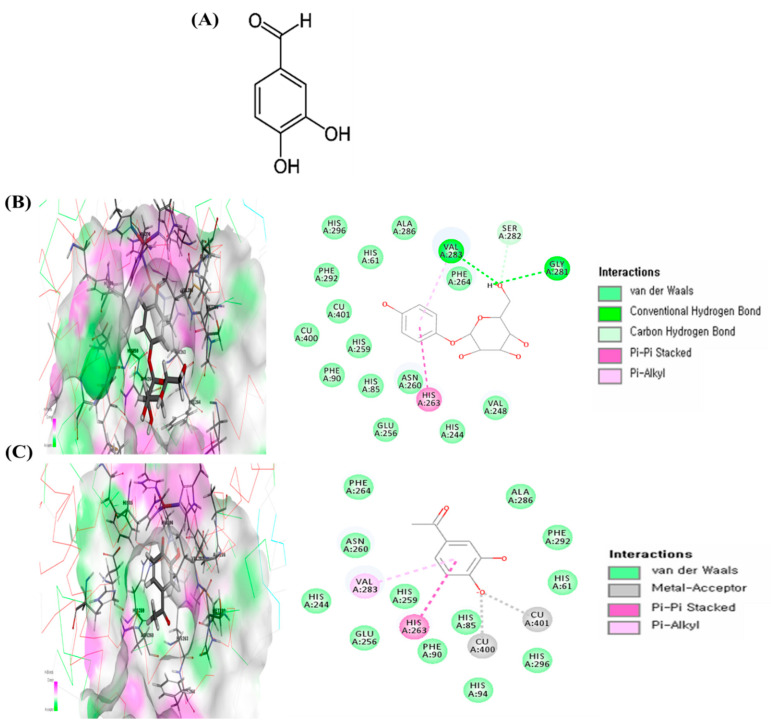
Chemical structure of protocatechuic aldehyde (PA) (**A**) and the specific interactions between PA and tyrosinase after automated docking of PA to the tyrosinase enzyme binding site. Predicted 3D structure of the tyrosinase (Protein Data Bank; PDB code: 2Y9X)–arbutin complex and 2D diagram (**B**). Predicted 3D structure of the tyrosinase (PDB 2Y9X)–PA complex and 2D diagram (**C**). Binding energy values were obtained from the Discovery Studio (DS) 3.0 binding energy calculation program.

**Figure 2 ijms-22-03861-f002:**
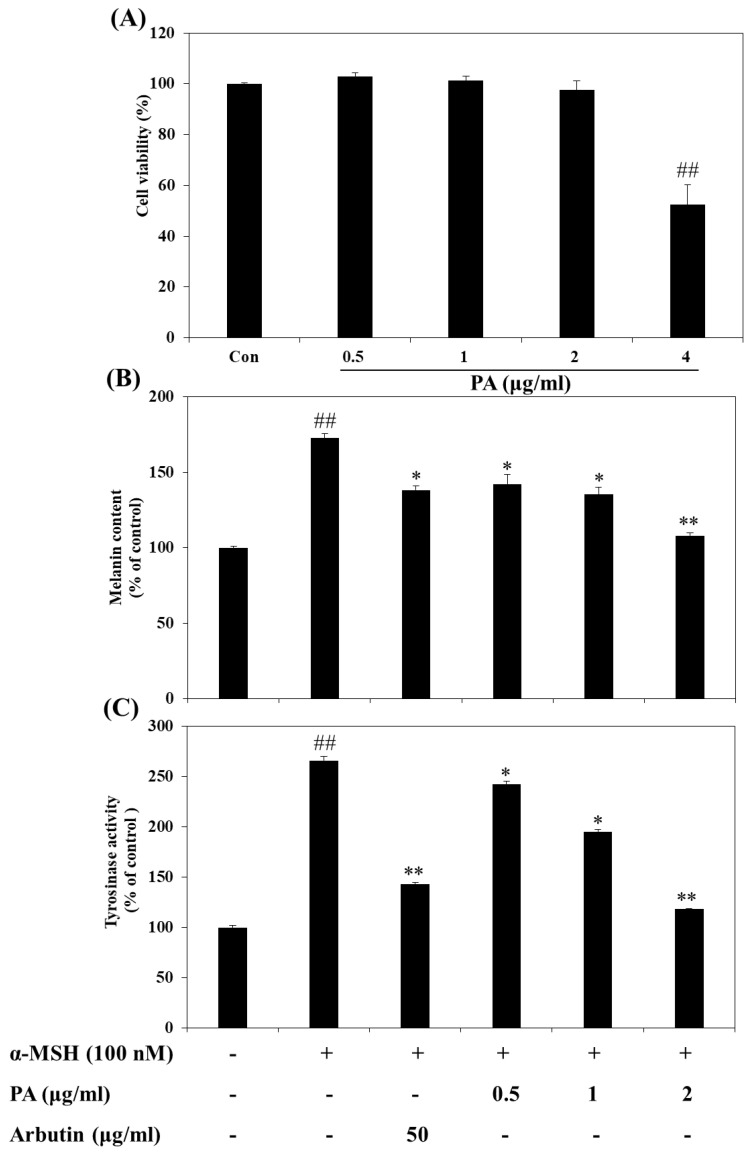
Effects of PA on cellular melanin synthesis and intracellular tyrosinase activity in α-melanocyte stimulating hormone (α-MSH)-stimulated B16F10 cells. Cells were treated with the indicated concentrations of PA for 72 h and then cell viability was determined by the 3-(4,5-dimethylthiazol-2-yl)-2,5-diphenyltetrazolium bromide (MTT) assay (**A**). The relative cellular melanin content (**B**) and intracellular tyrosinase activity (**C**) were measured at 72 h after treatment. Cells were exposed to 100 nM α-MSH in the presence of PA at the indicated concentrations or 50 µg/mL arbutin. Values are expressed as means ± SDs of triplicate experiments (*n* = 3). Note: ^##^
*p* < 0.01 compared to the untreated control group; * *p* < 0.05 and ** *p* < 0.01 compared to the α-MSH only group.

**Figure 3 ijms-22-03861-f003:**
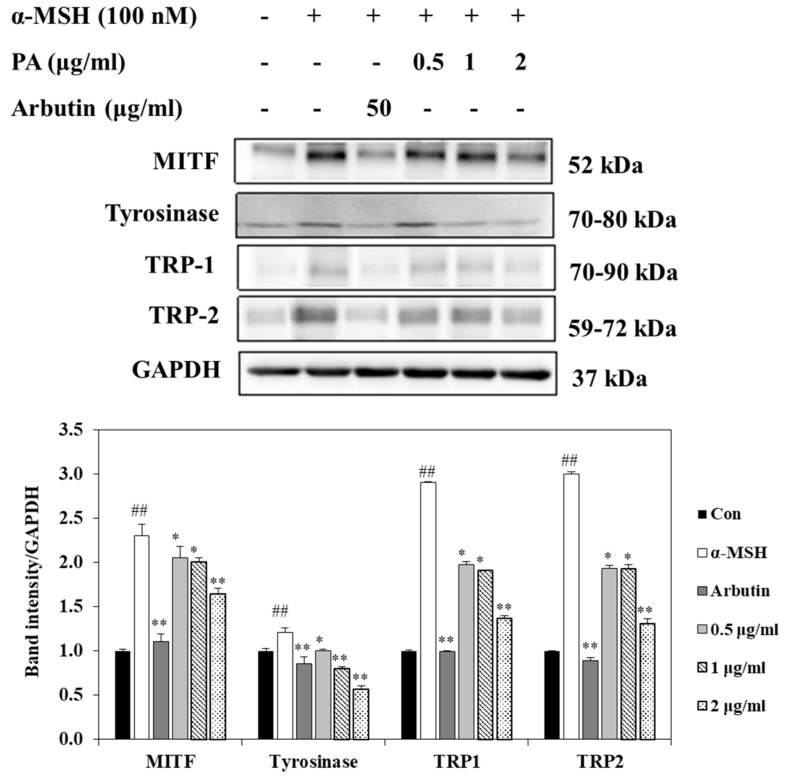
Inhibitory effects of PA on the expression levels of proteins related to melanogenesis in α-MSH-stimulated B16F10 cells. Cells were exposed to 100 nM α-MSH in the presence of PA at the indicated concentrations or 50 µg/mL arbutin. The expression levels of MITF, tyrosinase, TRP-1, and TRP-2 were measured using Western blot analysis and quantified using Image J. Values are expressed as means ± SDs of triplicate experiments (*n* = 3). Note: ^##^
*p* < 0.01 compared to the untreated control group; * *p* < 0.05 and ** *p* < 0.01 compared to the α-MSH group.

**Figure 4 ijms-22-03861-f004:**
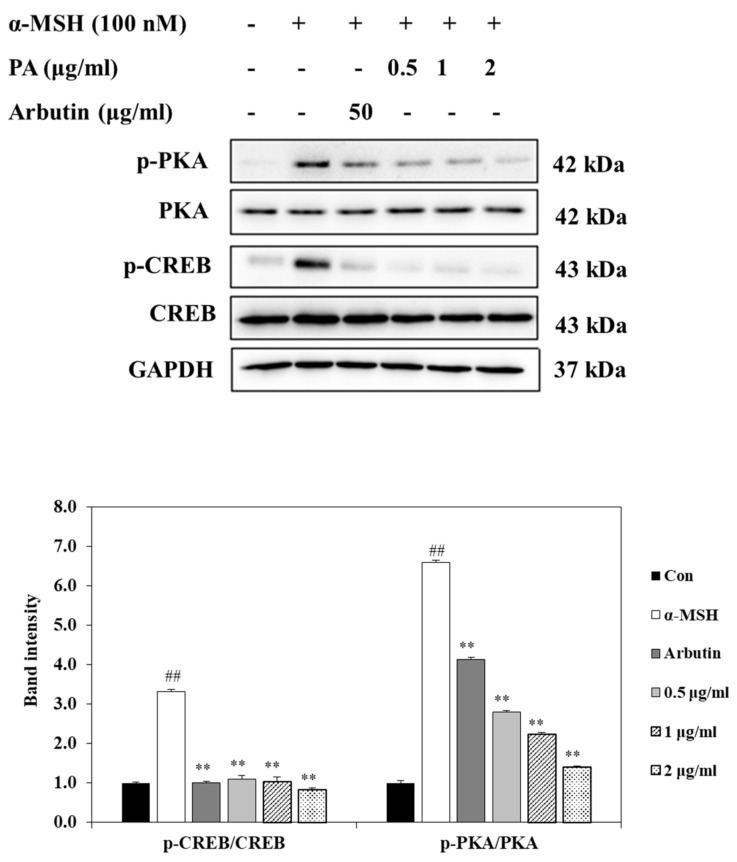
PA suppresses the PKA and CREB signaling pathways in α-MSH-stimulated B16F10 cells. Cells were exposed to 100 nM α-MSH in the presence of PA at the indicated concentrations or 50 µg/mL arbutin. The protein expression levels of p-PKA/PKA and p-CREB/CREB were determined using Western blot analysis and quantified using Image J. Values are expressed as means ± SDs of triplicate experiments (*n* = 3). Note: ^##^
*p* < 0.01 compared to the untreated control group; ** *p* < 0.01 compared to α-MSH group.

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
