# Peer review of "Protocatechuic Aldehyde Inhibits α-MSH-Induced Melanogenesis in B16F10 Melanoma Cells via PKA/CREB-Associated MITF Downregulation"

_ijms, 2021, doi:10.3390/ijms22083861_

Round 1
Reviewer 1 Report
Although the topic is of interest, the paper has major problems.
Experiments were performed on murine (B16) melanoma line, which has significant limitations Repeat of crucial experiments on human melanocytes is required.
The docking experiments were done on tyrosinase from Agaricus bisporus. Such in silico analyses should be done on mammalian tyrosinase instead.
Western blots for tyrosinase and TRP1 and TRP2 should show entire blot with indication of MW.
Nomenclature for abbreviations is incorrect when relating to murine melanogenesis related genes and proteins.
The authors do not know literature on melanin pigmentation and the references are randomly cited being not representative.
Example is for tyrosinase, where they should cite representative significant reviews for example by Hearing or Pawelek.
They should cite references following recommendations published in Endocrinology (Endocrinology, January 2010, 151(1):1–3) There are also many other issues with this manuscript
In the introduction the readers would appreciate information on hormonal regulation of melanin pigmentation (Physiol Rev 84, 1155-1228, 2000) and precursors to melanin (Pigment Cell Melanoma Res 25, 14-27, 2012).
In fact, based on the structure of the compounds, its in silico and experimental characteristics, one would suspect that it could counter-act stimulation of melanogenesis by L-tyrosine and L-dopa (J Cell Sci 89, 287-296, 198; Pigment Cell Res 2. 109-116, 1989; . Mol Cell Endocrinol 99, C7-C11, 1994). This should be discussed also with theoretical implications (J Theor Biol 143, 123-138, 1990)
Author Response
Dear Reviewer,
We thank you one more time and also the reviewer for the excellent comments and the valuable help. We have considered the comments of the reviewer and modified our manuscript. We hope the following responses and the corresponding revision of the manuscript fulfills the reviewer requirements for considering this manuscript for publication in IJMS.
Although the topic is of interest, the paper has major problems. Experiments were performed on murine (B16) melanoma line, which has significant limitations Repeat of crucial experiments on human melanocytes is required. The docking experiments were done on tyrosinase from Agaricus bisporus. Such in silico analyses should be done on mammalian tyrosinase instead.
Response: Thank you for very valuable comment and I agree your comment. However, as you know the 3D crystal structure of human tyrosinase is not available in PDB. Therefore, a homology modeling approach use to predict 3D structure of human tyrosinase. Unfortunately, Our in silico molecular docking analysis system can not do the homology modeling work. The mushroom tyrosinase (PDB code: 2Y9X) of Agaricus bisporus used in this study docking analysis is a major enzyme with high similarity and homology compared to human tyrosinase. Due to the above-mentioned good properties, the structural, functional, and biochemical characteristics of mushroom tyrosinase have been studied extensively as a model system for screening of tyrosinase inhibitors and melanogenic studies, enzyme-catalyzed reactions, and enzyme-inhibitor. For this reason, we also chose tyrosinase from Agaricus bisporus (PDB code: 2Y9X) because it is suitable for the examination of a model system for screening of tyrosinase inhibitors. I wish you to consider these points.
Western blots for tyrosinase and TRP1 and TRP2 should show entire blot with indication of MW.
Response: Thanks for your comment. The MW of all Western blots data have displayed and modified in the revised manuscript.
Nomenclature for abbreviations is incorrect when relating to murine melanogenesis related genes and proteins.
Response: Thanks for your comment. We have rechecked and revised abbreviations of murine melanogenesis related genes and proteins.
The authors do not know literature on melanin pigmentation and the references are randomly cited being not representative. Example is for tyrosinase, where they should cite representative significant reviews for example by Hearing or Pawelek. They should cite references following recommendations published in Endocrinology (Endocrinology, January 2010, 151(1):1–3) There are also many other issues with this manuscript In the introduction the readers would appreciate information on hormonal regulation of melanin pigmentation (Physiol Rev 84, 1155-1228, 2000) and precursors to melanin (Pigment Cell Melanoma Res 25, 14-27, 2012). In fact, based on the structure of the compounds, its in silico and experimental characteristics, one would suspect that it could counter-act stimulation of melanogenesis by L-tyrosine and L-dopa (J Cell Sci 89, 287-296, 198; Pigment Cell Res 2. 109-116, 1989; . Mol Cell Endocrinol 99, C7-C11, 1994). This should be discussed also with theoretical implications (J Theor Biol 143, 123-138, 1990)
Response: We thank you for this comment and completely agree with you. According to your comment, we have had it rechecked. The cite references were corrected again and the discussion section has been rewritten in more detail in the revised manuscript.
Reviewer 2 Report
In this study Ko and Lee have examined melanogenesis inhibition by protocatechuic aldehyde (PA) and mechanism of its action. They have studied molecular docking with (mushroom) tyrosinase, inhibition of melanin production and tyrosinase, expression levels of melanogenesis-related proteins, and phosphorylation of PKA and CREB. It thus appears that the study provided enough data to convince us that PA could be used as a whitening agent. However, there are two major flaws that suggest their conclusion questionable.
Major points:
- Efficacy as a tyrosinase inhibitor is not sufficient: PA is cytotoxic at 4 ug/mL (14 umol/L), indicating a high cytotoxicity. On the other hand, inhibition of melanin synthesis and tyrosinase activity are not strong enough (although significant) at 2 ug/mL.
- The use of mushroom tyrosinase to study is not appropriate: In fact, human tyrosinase often behaves quite differently from mushroom tyrosinase (Mann et al., 2018a, 2018b; Roulier et al., 2020; see below). Roulier et al. (2020) mention mushroom tyrosinase as a deceptive model. Thus, human tyrosinase has to be targeted in searching for effective whitening agents in these days.
Mann T, et al. (2018a). Inhibition of human tyrosinase requires molecular motifs distinctively different from mushroom tyrosinase. J Invest Dermatol 138, 1601-1608.
Man T, et al. (2018b). Structure-activity relationships of thiazolyl resorcinols, potent and selective inhibitors of human tyrosionase. Int J Mol Sci 19: 690 (review).
Roulier B, et a. (2020). Advances in the design of genuine human tyrosinase inhibitors for targeting melanogenesis and related pigmentations. J Med Chem 63: 13428-13443 (review).
Author Response
Dear Reviewer,
We thank you one more time and also the reviewer for the excellent comments and the valuable help. We have considered the comments of the reviewer and modified our manuscript. We hope the following responses and the corresponding revision of the manuscript fulfills the reviewer requirements for considering this manuscript for publication in IJMS.
In this study Ko and Lee have examined melanogenesis inhibition by protocatechuic aldehyde (PA) and mechanism of its action. They have studied molecular docking with (mushroom) tyrosinase, inhibition of melanin production and tyrosinase, expression levels of melanogenesis-related proteins, and phosphorylation of PKA and CREB. It thus appears that the study provided enough data to convince us that PA could be used as a whitening agent. However, there are two major flaws that suggest their conclusion questionable.
Major points:
- Efficacy as a tyrosinase inhibitor is not sufficient: PA is cytotoxic at 4 ug/mL (14 umol/L), indicating a high cytotoxicity. On the other hand, inhibition of melanin synthesis and tyrosinase activity are not strong enough (although significant) at 2 ug/mL.
Response: Thanks for your comment. Although high cytotoxicity was observed at 4 µg/ml concentration of PA, Especially, PA had similar inhibitory effect against tyrosinase activity and melanin synthesis than arbutin (50 µg/ml) at the low concentrations (2 µg/ml), which were evidenced stronger inhibitory effect than was observed with arbutin, a well-known melanogenesis inhibitor. Therefore, this evidence suggests that PA has potential as a skin whitening agent.
- The use of mushroom tyrosinase to study is not appropriate: In fact, human tyrosinase often behaves quite differently from mushroom tyrosinase (Mann et al., 2018a, 2018b; Roulier et al., 2020; see below). Roulier et al. (2020) mention mushroom tyrosinase as a deceptive model. Thus, human tyrosinase has to be targeted in searching for effective whitening agents in these days.
Mann T, et al. (2018a). Inhibition of human tyrosinase requires molecular motifs distinctively different from mushroom tyrosinase. J Invest Dermatol 138, 1601-1608.
Man T, et al. (2018b). Structure-activity relationships of thiazolyl resorcinols, potent and selective inhibitors of human tyrosionase. Int J Mol Sci 19: 690 (review).
Roulier B, et a. (2020). Advances in the design of genuine human tyrosinase inhibitors for targeting melanogenesis and related pigmentations. J Med Chem 63: 13428-13443 (review).
Response: Thank you for very valuable comment and I agree your comment. However, as you know the 3D crystal structure of human tyrosinase is not available in PDB. Therefore, a homology modeling approach use to predict 3D structure of human tyrosinase. Unfortunately, Our in silico molecular docking analysis system can not do the homology modeling work. The mushroom tyrosinase (PDB code: 2Y9X) of Agaricus bisporus used in this study docking analysis is a major enzyme with high similarity and homology compared to human tyrosinase. Due to the above-mentioned good properties, the structural, functional, and biochemical characteristics of mushroom tyrosinase have been studied extensively as a model system for screening of tyrosinase inhibitors and melanogenic studies, enzyme-catalyzed reactions, and enzyme-inhibitor. For this reason, we also chose tyrosinase from Agaricus bisporus (PDB code: 2Y9X) because it is suitable for the examination of a model system for screening of tyrosinase inhibitors. I wish you to consider these points.
Reviewer 3 Report
The study by Ko et al aims at investigation the role of Protocatechuic aldehyde in inhibition of α-MSH-induced melanogenesis in B16F10 melanoma cells via PKA/CREB–associated MITF downregulation.
Specific comments:
- Please improve the quality (resolution) of the figures.
- Fig. 3 - please specify which isoform of MITF is shown, and provide molecular weight markers for all blots.
- Please state "n" for all experiments.
- It would be valuable to discuss potential consequences/application of PA in induction of dedifferentiation in melanoma.
- 4.7. - please state the manufacturer of antibodies.
- Is PA cytotoxic against normal cells? This issue should be discussed, or (preferentially) investigated.
Author Response
Dear Reviewer,
We thank you one more time and also the reviewer for the excellent comments and the valuable help. We have considered the comments of the reviewer and modified our manuscript. We hope the following responses and the corresponding revision of the manuscript fulfills the reviewer requirements for considering this manuscript for publication in IJMS.
The study by Ko et al aims at investigation the role of Protocatechuic aldehyde in inhibition of α-MSH-induced melanogenesis in B16F10 melanoma cells via PKA/CREB–associated MITF downregulation.
Specific comments:
- Please improve the quality (resolution) of the figures.
Response: Thanks for your comment. We have improved quality of figures.
- Fig. 3 - please specify which isoform of MITF is shown, and provide molecular weight markers for all blots.
Response: Thanks for your comment. We have added MW markers for all blots.
- Please state "n" for all experiments.
Response: Thanks for your comment. We have added "n" for all experiments.
- It would be valuable to discuss potential consequences/application of PA in induction of dedifferentiation in melanoma.
Response: Thanks for your comment. We have included supportive text and added more evidence information in the discussion section in accordance with your comments.
- 4.7. - please state the manufacturer of antibodies.
Response: Thanks for your comment. We have added manufacturer of antibodies.
- Is PA cytotoxic against normal cells? This issue should be discussed, or (preferentially) investigated.
Response: Although high cytotoxicity was observed at 4 µg/ml concentration, previous study have reported that PA did not significantly affect cell viability at concentrations above 4 µg/ml in other cell line. Moreover, cytotoxicity protective effects against toxic substances were also reported.
Round 2
Reviewer 1 Report
The authors introduced some of the requested but not all corrections as recommended in the critique. Therefore, revisions are still required
As relates of choosing of non-mammalian tyrosinase for modeling, limitation of this model should be briefly mentioned. The same for using B16 melanoma.
I am surprised that the authors ignored the recommendations on "n fact, based on the structure of the compounds, its in silico and experimental characteristics, one would suspect that it could counter-act stimulation of melanogenesis by L-tyrosine and L-dopa" with proper referrals as listed.
Author Response
Responses to the reviewers’ comments on the manuscript and RESUBMISSION OF THE REVISED MANUSCRIPT
Manuscript ID: ijms-1155455
Title: Protocatechuic aldehyde inhibits α-MSH-induced melanogenesis in B16F10 melanoma cells via PKA/CREB–associated MITF downregulation
Dear Reviewer,
We thank you one more time and also the reviewer for the excellent comments and the valuable help. We have considered the comments of the reviewer and modified our manuscript. We hope the following responses and the corresponding revision of the manuscript fulfills the editor requirements for considering this manuscript for publication in IJMS.
The authors introduced some of the requested but not all corrections as recommended in the critique. Therefore, revisions are still required. As relates of choosing of non-mammalian tyrosinase for modeling, limitation of this model should be briefly mentioned. The same for using B16 melanoma.
Response: We thank you for this comment and completely agree with you. According to your comment, the discussion and conclusion section has been rewritten in more detail about the limitation of our model in the revised manuscript as below.
Human tyrosinase has significant role in the melanin biosynthesis in human body. Therefore, human tyrosinase is a potent target enzyme to design novel inhibitors for hyperpigmentation. However, the mushroom tyrosinase of Agaricus bisporus were used in this study docking analysis due to the 3D crystal structure of human tyrosinase is not available in PDB. Tyrosinase from Agaricus bisporus is a major enzyme with high similarity and homology compared to human tyrosinase. In addition, among non-mammalian tyrosinase, the tyrosinase from Agaricus bisporus have been studied extensively as a model system for screening of tyrosinase inhibitors and melanogenic studies due to good properties, the structural, functional, and biochemical characteristics. For this reason, we also was selected tyrosinase from Agaricus bisporus (PDB code: 2Y9X) because it is the most suitable for the examination of a model system for screening of tyrosinase inhibitors among non-mammalian tyrosinase.
Ref:
Zolghadri, S.; Bahrami, A.; Hassan Khan, M.T.; Munoz-Munoz, J.; Garcia-Molina, F.; Garcia-Canovas, F.; Saboury, A.A. A comprehensive review on tyrosinase inhibitors. J. Enzyme Inhib. Med. Chem. 2019, 34, 279–309.
Ito, S.; Wakamatsu, K. A convenient screening method to differentiate phenolic skin whitening tyrosinase inhibitors from leukoderma-inducing phenols. J. Dermatol. Sci. 2015, 80, 18–24.
However, it should be cautioned that tyrosinase from mushroom is different from human tyrosianse due to the presence of additional amino acids in the hydrophobic substrate-binding pocket of mushroom tyrosinase. Therefore, recently, a rather large number of studies have been directed to the utilization of human tyrosinase for the search and assessment of tyrosinase inhibitors. Although further human tyrosinase studies are required to clarify efficacy of PA in a tyrosinase activity inhibition, the present results suggest that PA could serve as a potential inhibitor of tyrosinase.
Ref:
Mann, T.; Scherner, C.; Röhm, K.H.; Kolbe, L. Structure-activity relationships of thiazolyl resorcinols, potent and selective inhibitors of human tyrosinase. Int J Mol Sci. 2018, 19, 690.
Roulier, B.; Pérès, B.; Haudecoeur, R. Advances in the design of genuine human tyrosinase inhibitors for targeting melanogenesis and related pigmentations. J. Med. Chem. 2020. 63, 13428-13443.
Although more extensive and further studies, including human tyrosinase and human melanoma cells studies, are needed for a thorough understanding of the potential inhibitor of PA on melanogenesis, our results confirm that PA could be explored as a possible skin lightening agent.
I am surprised that the authors ignored the recommendations on "n fact, based on the structure of the compounds, its in silico and experimental characteristics, one would suspect that it could counter-act stimulation of melanogenesis by L-tyrosine and L-dopa" with proper referrals as listed.
Response: We thank you for this comment and completely agree with you. According to your comment, the discussion section has been rewritten in more detail about the limitation of our model in the revised manuscript as below.
L-tyrosine and L-DOPA as the consecutive key substrates and intermediates of melanogenic pathway plays a important role as positive regulators of melanogenesis. In the melanogenic initial pathway, tyrosinase plays a central role in melanin biosynthesis by catalyzing the hydroxylation of tyrosine to form L-DOPA, and L-DOPA to DOPA quinone. Previous studies and present results showed that PA had a strong inhibitory effect in the tyrosinase activity inhibition experiment using dopa and tyrosine as substrates, respectively. Therefore, the results of previous and present studies clearly demonstrated that PA could counter-act stimulation of melanogenesis by both L-tyrosine and L-DOPA.
Ref:
Slominski, A.; Zmijewski, M.A.; Pawelek, J. L-tyrosinase and L-dihydroxyphenylalanine as hormone-like regulators of melanocyte functions. Pigment Cell Melanoma Res. 2012, 25, 14-27.
No, J.K.; Kim, M.S.; Kim, Y.J.; Bae, S.J.; Choi, J.S.; Chung, H.Y. Inhibition of tyrosinase by protocatechuic aldehyde. Am J Chin Med. 2004, 32, 97-103.
Reviewer 2 Report
In my previous review I mentioned that the use of mushroom tyrosinase to study melanogenesis inhibitors is not appropriate. This suggestion should not be ignored and still needs to be discussed in the Discussion section by citing Mann et al. (2018) Structure-activity relationships of thiazolyl resorcinol, potent and selective inhibitors of human tyrosinase. Int. J. Mol. Sci. 19: 690 and Roulier B. et al. (2020) Advances in the design of genuine human tyrosinase inhibitors for targeting melanogenesis and related pigmentations. J. Med. Chem. 63: 13428-13442. The reviewer believes that the use of human tyrosinase (at least mammalian tyrosinase) is essential in the future studies searching for whitening agents.
One positive point of this study that I did not mention in the previous study is the following. Protocatechuic aldehyde belongs to catechols which usually act as substrates for tyrosinase. So some readers wonder why this catholic compound is not oxidized by (mushroom) tyrosinase. How about protocatechuic acid? The reviewer believes that those catechols with an electron-withdrawing group attached directly to the benzene ring do not act as substrates for tyrosinase but might act as inhibitors as shown in the present study. This aspect should also be discussed by citing Ito and Wakamatsu (2015) A convenient screening method to differentiate phenolic skin whitening tyrosinase inhibitors from leukoderma-inducing phenols. J. Dermatol. Sci. 80: 18-24. The reviewer wonders whether the authors have confirmed that protocatechuic aldehyde does not act as a substrate (not inhibitor) for mushroom tyrosinase.
Author Response
Responses to the reviewers’ comments on the manuscript and RESUBMISSION OF THE REVISED MANUSCRIPT
Manuscript ID: ijms-1155455
Title: Protocatechuic aldehyde inhibits α-MSH-induced melanogenesis in B16F10 melanoma cells via PKA/CREB–associated MITF downregulation
Dear Reviewer,
We thank you one more time and also the reviewer for the excellent comments and the valuable help. We have considered the comments of the reviewer and modified our manuscript. We hope the following responses and the corresponding revision of the manuscript fulfills the editor requirements for considering this manuscript for publication in IJMS.
In my previous review I mentioned that the use of mushroom tyrosinase to study melanogenesis inhibitors is not appropriate. This suggestion should not be ignored and still needs to be discussed in the Discussion section by citing Mann et al. (2018) Structure-activity relationships of thiazolyl resorcinol, potent and selective inhibitors of human tyrosinase. Int. J. Mol. Sci. 19: 690 and Roulier B. et al. (2020) Advances in the design of genuine human tyrosinase inhibitors for targeting melanogenesis and related pigmentations. J. Med. Chem. 63: 13428-13442. The reviewer believes that the use of human tyrosinase (at least mammalian tyrosinase) is essential in the future studies searching for whitening agents.
Response: We thank you for this comment and completely agree with you. According to your comment, the discussion and conclusion section has been rewritten in more detail about the limitation of our model in the revised manuscript as below.
Human tyrosinase has a significant role in melanin biosynthesis in the human body. Therefore, human tyrosinase is a potent target enzyme to design novel inhibitors for hyperpigmentation. However, the mushroom tyrosinase of Agaricus bisporus were used in this study docking analysis due to the 3D crystal structure of human tyrosinase is not available in PDB. Tyrosinase from Agaricus bisporus is a major enzyme with high similarity and homology compared to human tyrosinase. In addition, among non-mammalian tyrosinase, the tyrosinase from Agaricus bisporus have been studied extensively as a model system for screening of tyrosinase inhibitors and melanogenic studies due to its good properties, the structural, functional, and biochemical characteristics. For this reason, we also were selected tyrosinase from Agaricus bisporus (PDB code: 2Y9X) because it is the most suitable for the examination of a model system for screening of tyrosinase inhibitors among non-mammalian tyrosinase.
Ref:
Zolghadri, S.; Bahrami, A.; Hassan Khan, M.T.; Munoz-Munoz, J.; Garcia-Molina, F.; Garcia-Canovas, F.; Saboury, A.A. A comprehensive review on tyrosinase inhibitors. J. Enzyme Inhib. Med. Chem. 2019, 34, 279–309.
Ito, S.; Wakamatsu, K. A convenient screening method to differentiate phenolic skin whitening tyrosinase inhibitors from leukoderma-inducing phenols. J. Dermatol. Sci. 2015, 80, 18–24.
However, it should be cautioned that tyrosinase from mushroom is different from human tyrosinase due to the presence of additional amino acids in the hydrophobic substrate-binding pocket of mushroom tyrosinase. Therefore, recently, a rather large number of studies have been directed to the utilization of human tyrosinase for the search and assessment of tyrosinase inhibitors. Although further human tyrosinase studies are required to clarify the efficacy of PA in a tyrosinase activity inhibition, the present results suggest that PA could serve as a potential inhibitor of tyrosinase.
Ref:
Mann, T.; Scherner, C.; Röhm, K.H.; Kolbe, L. Structure-activity relationships of thiazolyl resorcinols, potent and selective inhibitors of human tyrosinase. Int J Mol Sci. 2018, 19, 690.
Roulier, B.; Pérès, B.; Haudecoeur, R. Advances in the design of genuine human tyrosinase inhibitors for targeting melanogenesis and related pigmentations. J. Med. Chem. 2020. 63, 13428-13443.
Although more extensive and further studies, including human tyrosinase and human melanoma cells studies, are needed for a thorough understanding of the potential inhibitor of PA on melanogenesis, our results confirm that PA could be explored as a possible skin-lightening agent.
One positive point of this study that I did not mention in the previous study is the following. Protocatechuic aldehyde belongs to catechols which usually act as substrates for tyrosinase. So some readers wonder why this catholic compound is not oxidized by (mushroom) tyrosinase. How about protocatechuic acid? The reviewer believes that those catechols with an electron-withdrawing group attached directly to the benzene ring do not act as substrates for tyrosinase but might act as inhibitors as shown in the present study. This aspect should also be discussed by citing Ito and Wakamatsu (2015) A convenient screening method to differentiate phenolic skin whitening tyrosinase inhibitors from leukoderma-inducing phenols. J. Dermatol. Sci. 80: 18-24. The reviewer wonders whether the authors have confirmed that protocatechuic aldehyde does not act as a substrate (not inhibitor) for mushroom tyrosinase.
Response: We thank you for this comment and completely agree with you. According to your comment, the discussion section has been rewritten in more detail about the limitation of our model in the revised manuscript as below.
Substrates for tyrosinase are generally either phenols, which undergo monooxygenation, or catechols, which undergo oxidation, leading in both cases to ortho-quinones. However, it has previously been reported that some polyphenolic compounds with an electron-withdrawing group attached directly to the benzene such as hydroquinone and catechols do not act as substrates in mushroom tyrosinase. Thus, we believe that PA, naturally derived catechol, with an electron-withdrawing group attached directly to the benzene ring does not act as substrates for tyrosinase but might act as an inhibitor as shown in the present study. Furthermore, it is necessary to further examine the efficacy of PA by checking whether or not the PA acts as a substrate of human tyrosinase being modified on exposure to the human tyrosinase.
Ref:
Pillaiyar, T.; Manickam, M.; Namasivayam, V. Skin whitening agents: Medicinal chemistry perspective of tyrosinase inhibitors. J. Enzyme Inhib. Med. Chem. 2017, 32, 403–425.
Ito, S.; Wakamatsu, K. A convenient screening method to differentiate phenolic skin whitening tyrosinase inhibitors from leukoderma-inducing phenols. J. Dermatol. Sci. 2015, 80, 18–24.
Reviewer 3 Report
All comments have been addressed in a satisfactory way.
Author Response
We thank you one more time and also the reviewer for the excellent comments and the valuable help.
Round 3
Reviewer 1 Report
For the most part the authors adequately replied.
There is one minor correction pending; the fragment on tyrosine and dopa will benefit by addition of original citations ((J Cell Sci 89, 287-296, 1988; Pigment Cell Res 2. 109-116, 1989) in addition to already cited review
Author Response
For the most part the authors adequately replied. There is one minor correction pending; the fragment on tyrosine and dopa will benefit by addition of original citations ((J Cell Sci 89, 287-296, 1988; Pigment Cell Res 2. 109-116, 1989) in addition to already cited review
Response: Thanks for your comment. We have added citations in accordance with your comments as below.
- Slominski, A.; Moellmann, G.; Kuklinska, E.; Bomirski, A.; Pawelek, J. Positive regulation of melanin pigmentation by two key substrates of the melanogenic pathway, L-tyrosine and L-dopa. J. Cell Sci. 1988, 89, 287-296.
- Slominski, A.; Moellmann, G.; Kuklinska, E. L‐tyrosine, L‐dopa, and tyrosinase as positive regulators of the subcellular apparatus of melanogenesis in Bomirski Ab amelanotic melanoma cells. Pigm. Cell. Res. 1989, 2, 109-116.
This manuscript is a resubmission of an earlier submission. The following is a list of the peer review reports and author responses from that submission.